# Comparatively Evolution and Expression Analysis of GRF Transcription Factor Genes in Seven Plant Species

**DOI:** 10.3390/plants12152790

**Published:** 2023-07-27

**Authors:** Zhihan Cheng, Shiqi Wen, Yuke Wu, Lina Shang, Lin Wu, Dianqiu Lyu, Hongtao Yu, Jichun Wang, Hongju Jian

**Affiliations:** 1Integrative Science Center of Germplasm Creation in Western China (CHONGQING) Science City, Southwest University, Chongqing 400715, China; czh010812@163.com (Z.C.); carnival@email.swu.edu.cn (S.W.); winkwu730@foxmail.com (Y.W.); cxldshanglina@163.com (L.S.); wulin2022@swu.edu.cn (L.W.); smallpotatoes@126.com (D.L.); wjchun@swu.edu.cn (J.W.); 2College of Agronomy and Biotechnology, Southwest University, Chongqing 400715, China; 3State Cultivation Base of Crop Stress Biology for Southern Mountainous Land of Southwest University, Chongqing 400715, China; 4Chongqing Key Laboratory of Biology and Genetic Breeding for Tuber and Root Crops, Chongqing 400715, China; 5Suihua Branch of Heilongjiang Academy of Agriculture Sciences, Suihua 152052, China; yuhongtao9900@163.com

**Keywords:** GRF, abiotic stress, expression pattern, evolution, GIF, interaction

## Abstract

Growth regulatory factors (GRF) are plant-specific transcription factors that play pivotal roles in growth and various abiotic stresses regulation. However, adaptive evolution of GRF gene family in land plants are still being elucidated. Here, we performed the evolutionary and expression analysis of GRF gene family from seven representative species. Extensive phylogenetic analyses and gene structure analysis revealed that the number of genes, QLQ domain and WRC domain identified in higher plants was significantly greater than those identified in lower plants. Besides, dispersed duplication and WGD/segmental duplication effectively promoted expansion of the GRF gene family. The expression patterns of GRF gene family and target genes were found in multiple floral organs and abundant in actively growing tissues. They were also found to be particularly expressed in response to various abiotic stresses, with stress-related elements in promoters, implying potential roles in floral development and abiotic stress. Our analysis in GRF gene family interaction network indicated the similar results that GRFs resist to abiotic stresses with the cooperation of other transcription factors like GIFs. This study provides insights into evolution in the GRF gene family, together with expression patterns valuable for future functional researches of plant abiotic stress biology.

## 1. Introduction

As sessile organisms, plants continually encounter adverse environments, such as diseases, pests, drought, extreme temperatures, high salt and heavy metals during their whole life cycles [1]. High temperature is one of the most important factors, having a significant impact on basic life activities [2]. High temperature stress causes harmful effects on plants, including denaturation of proteins, inhibition of carbohydrate transport, alteration of lipid membrane fluidity and integrity [3,4]. Globally, many plants are sensitive to high temperature stress, which ultimately affects the universal economy and crop production [4]. Plants have sensed and initiated spontaneous responses when confronted with heat stress, such as regulating heat stress enzymes and evolving thermo-tolerance factors [5,6]. Among them, GROWTH-REGULATING FACTORs (GRFs) represent a plant-specific TF family, which responds to high temperature stress and plays an important role in improving tolerance in thermal environment [7,8,9].

As a typical master regulators, GRFs were the first factors to be identified as a gibberellic acid (GA)-induced genes in intercalary meristem internodes from rice two decades ago [10]. Highly conserved functional domains QLQ (glutamine, leucine and glutamine) and WRC (tryptophan, arginine and cysteine) in the N-terminus, as well as a variable C-terminal domain were observed in GRF proteins [11]. To form a transcriptional co-activator, the QLQ domains combined with GRF-Interacting Factor (GIFs) existed in all eukaryotes [12]. QLQ domains in GRFs also present in the Switch/Sucrose Non-fermenting gene (SWI2/SNF2) protein to facilitate the interaction with other proteins by forming a complex related to chromatin remodeling [11]. While the WRC domains are plant-specific, which act as DNA binding agents and target intra-nuclear transcription factors to regulate downstream gene expression [13]. Though, low similarities among sequences, the C-terminal domains are essential for their transcriptional activity [14]. Transcription activity was lost in sequences with truncated C-terminal domains in the previous study [15]. Taken together, GRFs function as a small family of transcription factors that require both the N-terminal conserved WRC and QLQ domains and C-terminal diverse sequences.

Additionally, GRFs have been reported to have a wide range of biological functions and participate in a variety of metabolic pathways [16,17,18]. Functional and molecular analyses have shown that several *GRFs* play critical roles in plant development (such as flowering, cell proliferation, stem elongation, seed and root development) and stress response [7,19]. A recent study in Arabidopsis revealed that *Atgrf1/Atgrf2/Atgrf3* triplet mutants promoted to produce smaller leaves and cotyledons through modulating the cell expansion [20]. AtGRF5 works with AtGIF1 to positively regulate the development of leaf primordia [21]. In rice, *OsGRF4* is reported to increase spike length through managing the expression of cytokinin oxidase/dehydrogenase genes (*CKX1* and *CKX5*) [19]. In *Brassica napus*, *BnGRF2* raised oil yield by enlarging cell number in the seed and increasing the leaf photosynthesis [22]. Besides the functions in plant development, GRFs also act critical roles in response to abiotic stresses [14]. *AtGRF7* negatively regulates the tolerance to osmotic stress by inhibiting the expression of osmotic stress-responsive genes, including DEHYDRATION RESPONSIVE ELEMENT-BINDING PROTEIN2A (*DREB2A*) [23]. In *B. rapa*, *BrGRF5* positively regulates growth and development, while negatively responding to salt and osmotic stresses [24]. In addition, *ZmGRF4* and *ZmGRF13* are significantly induced by heat, salt and drought stresses, indicating that *ZmGRF* genes may act critical functions in response to these abiotic stresses in maize [25]. The expression levels of *GhGRF1a-At* and *GhGRF9b-Dt* are decreased under the exposure of cold and polyethylene glycol (PEG) stresses, while are increased after the heat and salt stresses, suggesting that GRF genes in cotton play different roles in response to various abiotic stresses [26]. In cassava, the *MeGRF4* is proved to be responsive to low temperature and salt stresses [27]. Both *FvGRF4* and *FvGRF9* were up-regulated under high temperature stress in strawberry [9]. These findings provide enough proof that GRFs act critical roles in plant development and stress response. 

In recent years, a growing body of plant species are being sequenced and this is very advantageous to detect GRF family members in the whole genome. Up to date, 9 *GRF* genes in *Arabidopsis thaliana* [28], 11 *GRF* genes in *Oryza sativa* [29], 11 *GRF* genes in *Camellia sinensis* [30], 9 *GRF* genes in *Ricinus communis* [31], 12 *GRF* genes in *Hordeum vulgare* [32], 8 *GRF* genes in *Sorghum bicolor* [33], 14 *GRF* genes in Zea mays [25], 10 *GRF* genes in *Setaria italica* [34], 19 *GRF* genes in *Panicum virgatum* [35] and 11 *GRF* genes in *Sesamum indicum* [36] were identified in previous studies. However, previous reports mainly provided evidence of GRFs gene origins and diversification from molecular structure and composition [13], the evolutionary patterns of the GRF gene family among multiple plant species, such as gene replication, selection pressure and expression modes are still unclear. With the continuous updating of genome sequencing technology, more and more plant genome information has been released, which makes this analysis possible. In this study, to reveal evolutionary relationships, we identified GRF gene members in seven plant species ranging from lower to higher species based on the Phytozome database: Thallophyte, Bryophyta, Pteridophyta, Monocots, Lauraceae, Rosales and Cruciferae. Phylogenetic tree, exon and intron structures of *GRF* genes, *GRFs* duplication and loss during evolution, syntenic gene pairs between seven representative plants, functional enrichment of upstream and downstream genes as well as the cis-acting regulatory elements in their promoter regions were analyzed. Furthermore, we also characterized their expression profiles in different tissues, developmental stages and their responses to various abiotic stresses among selected species. All these results make it possible to carry out comprehensive evolutionary conclusions, which extend the utility of GRFs as a foundational model transcription factor for in-depth study into gene biology and function research.

## 2. Results

### 2.1. Identification of GRF Gene Members in Seven Plants

To systematically study the evolution and expression of GRFs, 70 GRF genes from 7 species were identified using BlastP analysis in phytozome (Figure 1; Appendix A). To further precisely study the evolutionary relationships of the seven species, a phylogenetic tree of GRF genes was constructed based on 70 amino acid sequences from these species (Figure 1a). Four subfamilies were divided according to the characteristics of GRF domains (Figure 1; Appendix A). Nineteen GRF genes were detected in rice, while *O. lucimarinus* contains only one GRF member, suggesting that GRF family replicates and expands during evolution (Figure 1a①,②). Interestingly, the lowest ratio of gene numbers of GRF family numbers to the number of all genes in the plant genome and the highest ratio of the length of GRF family to the length of all genes were detected in *S. moellendorffii* (Figure 1a③,④), indicating that GRF genes in the bottom plants were not prone to gene loss during evolution. Besides *A. thaliana* and *S. moellendorffili*, the ratio of GRF length to the whole genome was all greater than 0.4 (Figure 1a⑤), indicating the similarity among members of the GRFs. There was an obvious boundary in the evolutionary tree that separates the protein sequences with complete domains (containing both WRC and QLQ domains) and the sequence with partial domains (containing either WRC or QLQ domain), while sequences containing partial domains accounted for 38.6%, mostly in *O. sativa* and *C. kanehirae* (Figure 1a⑥). We also found that the 14618 gene in *O. lucimarinus* had the most complete motifs but with only one WRC domain, indicating that GRF gene family may evolve new domains to play diverse functions (Figure 1a⑦). Besides QLQ and WRC domains, other six domains were identified in GRF proteins (Figure 1a⑧). Rmic and JmjC domains were detected in several GRF sequences of *C. kanehirae* and *F. vesca*, indicating that these GRF genes may evolve new functions. The JmjC domain-containing proteins mainly catalyzed the demethylation of histone lysine to maintain normal metabolism. 

To explore the original evolution and relationship of the GRF gene family in seven species, GRF motif analysis using the MEME website was performed (Figure 1b). Among them, motif 2, motif 4, motif 6 and motif 10 were the longest with 50 amino acid residues, followed by motif 8 and motif 9 with 38 amino acid residues (Appendix A). motif 9 is partially absent in all seven species (Figure 1b). In *O. lucimarinus*, *Selaginella moellendorffii*, *Fragaria vesca* and *Oryza sativa*, all 10 motifs were partially absent. Only 8 motifs in *M. polymorpha* and *A. thaliana* are partially absent. However, only 7 motifs were partially absent in *Cinnamomum kanehirae* (Figure 1b). This phenomenon indicated that GRF gene family had sequence divergence in the evolutionary process.

### 2.2. Cis-Acting Analysis in GRF Promoters

To further explore the regulated patterns of GRF genes in seven representative plants, the cis-acting elements in the promoter sequences (2 kb sequences upstream of ATG) of GRF family genes were predicted using the PLANTCARE database. In total, 55 types of cis-acting elements with 23 kinds of functions from 70 GRF promoter sequences were screened (Figure 2a, Appendix A). Apart from the light-responsive element (occurred 727 times), various hormone-responsive elements (239 meJA-responsiveness, 231 zein metabolism regulation, 187 abscisic acid responsiveness, 67 gibberellin-responsive elements, 51 salicylic acid responsiveness and 50 auxin responsiveness), stress-responsive elements (140 anaerobic induction, 54 low-temperature responsiveness, 48 drought inducibility, 42 defense and stress responsiveness), and other regulatory elements (36 meristem expression, 32 anoxic specific inducibility, 22 endosperm expression, 16 circadian control, 5 maximal elicitor-mediated activations, 4 cell cycle regulation, 4 seed-specific regulation, 3 flavonoid biosynthetic genes regulation, 2 wound-responsive elements, 1 phytochrome down-regulation expression and 1 differentiation of the palisade mesophyll cells) were detected (Figure 2, Appendix A). All these results suggested that GRF family genes in seven representative plants may be regulated by various hormones, environmental stresses and spatial development. Besides, the G-box and ABRE could be found in all these 70 GRF promoter sequences, suggesting that plants have light-regulating properties for maintaining *O. lucimarinus* metabolism and dehydration properties for defending against low temperature and drought stress. The AT-rich motif was found in all higher five representative species except *O. lucimarinus* and *M. polymorpha,* indicating that the lower plants may have chromatinization and transcriptional regulation of immunity in other ways. More than 15 cis-acting elements in each GRF gene promoter were detected except *Mapoly1353s0001.1*, which only contains eight cis-elements (Figure 2a, Appendix A). Among them, *LOC_Os02g45570.1* (in *O. sativa*) ranked first with 40 cis-acting elements, followed by *LOC_Os10g30890.1* (37) and 14618 (35) in *O. lucimarinus* (Figure 2a, Appendix A). Higher species have more cis-acting elements than lower species, indicating that species have generated more cis-acting elements during the evolution process. 

To further investigate the relationship of the promoter sequences in seven representative plants, a polygenetic tree was constructed using these 70 promoter sequences (Figure 2b). The promoter sequences of the seven representative species could be divided into seven groups, with the promoter sequences of the same species distributed in each group. The promoter sequences showed that plants have evolved similar key cis-acting elements of promoter region to perform efficient functions in response to harsh environmental stresses and growth demands. Compared with the polygenetic protein tree of seven representative species, the promoter sequences had a higher degree of variability.

### 2.3. Duplication and Loss of GRF Gene Family in Seven Typical Species

Gene duplication is a key process in species evolution, differentiation and diversity [37]. Here, we performed gene duplication and loss analysis of GRF gene family in seven typical species using Notoung software. There were 0, 1, 2, 5, 3, 3, and 0 genes duplicated and 0, 3, 4, 2, 1, 5, and 1 genes lost in *O. lucimarinus*, *M. polymorpha*, *S. moellendorffii*, *C. kanehirae*, *O. sativa*, *A. thaliana* and *F. vesca*, respectively (Figure 3a, Appendix A). Based on the phylogenetic time tree, in the lineages of the common ancestor of *C. kanehirae*, *O. lucimarinus*, *A. thaliana* and *F. vesca,* 8 genes were duplicated and 23 genes were lost. In the lineages of *M. polymorpha*, *S. moellendorffii*, *C. kanehirae*, *O. sativa*, *A. thaliana* and *F. vesca*, 23 genes were duplicated and 16 genes were lost (Figure 3a). We also observed that all the representative species underwent duplication and loss during evolution except *O. lucimarinus*, indicating that whole genome replication may not have occurred in lower plants.

We further detected five types of gene duplication, including singleton, dispersed, proximal, tandem, and WGD/segmental in seven representative plants (Figure 3b and Figure 5c). In all genes, the dispersed replication type accounted for the largest percentage, followed by singleton (Figure 3b). Besides, among the five types, dispersed and WGD/segmental counted the majority and played critical roles in GRF gene family expansion (Figure 3; Appendix A). Only one singleton type of GRF duplication was detected in *O. lucimarinus* and the type of dispersed in GRF genes counted 12.1%, 6.1%, 21.2%, 15.2%, 27.3% and 18.2% in *M. polymorpha*, *S. moellendorffii*, *C. kanehirae*, *A. thaliana*, *O. sativa* and *F. vesca*, respectively (Figure 3c; Appendix A). No GRF member belonging to WGD or Segmental was detected in *O. lucimarinus* and *M. polymorpha*, and the percentage of GRF genes belonging to WGD/segmental was 7.4%, 25.9%, 14.8%, 29.6% and 22.2% in *S. moellendorffii*, *C. kanehirae*, *A. thaliana*, *O. sativa* and *F. vesca*, respectively (Figure 3c). Several GRF genes belong to proximal and singleton (Figure 3c). Except for *O. sativa*, nearly no GRF genes belong to tandem (Figure 3c). Additionally, we found that more than 50% of the species were detected in the mesozoic era, while *O. lucimarinus*, *M. polymorpha* and *S. moellendorffii* existed in neo-proterozoic, lower palozoic and middle palozoic (Figure 3d), which indicated the evolutionary chronology.

### 2.4. The Evolution of Syntenic GRF Genes among Inter- and Intra- Representative Plants Was Accompanied by Genome-Wide Polyploidy Events

To further explore the relationship of the *GRF* family genes among the seven representative plants, we performed syntenic analysis. Nineteen orthologous gene pairs between *C. kanehirae* and *O. sativa*, 17 between *A. thaliana* and *F. vesca*, 9 between *F. vesca* and *O. sativa*, 7 between *C. kanehirae* and *F. vesca*, 5 between *C. kanehirae* and *A. thaliana* and 3 between *A. thaliana* and *O. sativa* were detected (Figure 4a; Appendix A). No orthologous gene pairs were detected among *O. lucimarinus, M. polymorpha* and *S. moellendorffii.* The number of orthologous genes between *C. kanehirae* and *O. sativa* was much more than the other comparisons, indicating that the relationship between them was closer than others (Appendix A). 

Furthermore, for judging the selective pressure acting on *GRFs* gene, nonsynonymous (Ka) and synonymous (Ks) substitutions with selected types of orthologous gene pairs were also calculated. On the whole, the values of Ka/Ks ratio were less than 1.0 among all the orthologous gene pairs in representative plants, suggesting that these GRF genes underwent purifying selection. 

### 2.5. GRFs May Participate in External Stimuli and Floral Development

Because of the incompleteness of many crop GRF member databases on the Internet, the model crop Arabidopsis was selected as the study object, and we intend to use it to identify potential abiotic stress-related candidate genes. To further explore the potential functions of GRFs, 582 upstream genes and 3679 downstream genes of GRFs in Arabidopsis were screened and used to construct the interaction network (Figure 5a; Appendix A). Additionally, the upstream and downstream genes of each GRF gene in Arabidopsis varied greatly. In detail, 39–376 upstream and 176–1857 downstream genes were screened from nine GRF genes in Arabidopsis (Figure 5b, Appendix A). Among them, 197 genes were commonly detected in upstream and downstream genes (Figure 5c; Appendix A). Noteworthy, 120 common genes encode transcription factors, which were belonging to 25 gene families (Figure 5d). Among them, MYB accounted for the highest percentage, followed by HB, bHLH, and MADS, indicating that these TF families may join together with GRFs to participate in biological processes.

To further explore the function of the target genes of *GRFs* in the network, KEGG enrichment analyses were performed with the parameter: *q*-value < 0.05. Ten pathways were significantly enriched in the downstream genes regulated by GRF in Arabidopsis (Figure 6; Appendix A). Most members of GRF downstream genes were associated in the metabolic pathways and secondary metabolic (Figure 6a). In addition, fatty acid biosynthesis, plant hormone signal transduction, glucosinolate biosynthesis and MAPK signaling pathway were also enriched (Figure 6a; Appendix A). As reported in previous studies, plant fatty acid metabolism participates in high temperature response [38]. In our study, 164 downstream GRF genes mainly involved in fatty acid-related pathways, such as biosynthesis, metabolism and degradation (Figure 6a; Appendix A). Taken together, these results suggested that the target genes regulated by GRFs participated in a wide range of essential and crucial pathways.

In addition, we also performed GO enrichment for downstream genes of GRF in Arabidopsis (Figure 6b). Most of the target genes were involved in responses to stimuli (GO:0050896), hormones (GO:0009725), abiotic stimuli (GO:0009628), floral organ development (GO:0048437), and flower development (GO:0009908) (Figure 6b; Appendix A), which suggested that GRFs may participate in the regulation of external stimuli and floral development. 

### 2.6. Exploring the Expression Patterns of GRF Gene Family under Diverse Conditions

To comprehensively gain insight into the functions of GRFs, we selected monocotyledon (*O. sativa*) and dicotyledon (*A. thaliana*), respectively and performed expression analysis of GRFs under diverse stresses and different tissues (Figure 7). In Arabidopsis, there were two *GRFs* expression values lost (*At3g52910* and *At2g45480*). *AtGRFs* had similar expression patterns under abiotic stresses (drought, heat, salt, oxidative, cold and osmotic) except *At4g24150,* which was greatly upregulated by heat, drought and salt stresses after 1 h in the root, followed by *At2g22840* after oxidative stresses after 3 h, 6 h, 12 h and 24 h in the shoot (Figure 7a; Appendix A). Besides, GRF genes showed tissue-specific expression patterns. *At2g22540, At5g13180* and *At3g6189* were preferentially expressed in the shoot apex (inflorescence, transition and vegetative), followed by flower stage (flower stage 9, flower stage 12 carpels and flower stage 15 carpels) (Figure 7b; Appendix A). Similarly, the *GRF* genes in rice also showed expression specificity in different tissues. Most of genes were found abundantly expressed in meristem, flower and embryo, while *LOC_OS07G28430, LOC_OS07G37140* and *LOC_OS03G22540* were upregulated by various abiotic stresses (Figure 7c; Appendix A). *LOC_OS02G28580* was the only one with significantly high expression under heat stress, particularly rising at flowering and endosperm (Figure 7c; Appendix A). Interestingly, compared with the replication gene *LOC_OS02G47280*, *LOC_OS04G51190* highly expressed under cold and shade conditions (Figure 7c; Appendix A). In general, most of the *GRF* genes were more highly expressed in vegetative tissues and flower organs compared to developing tissues, suggesting that they may play an important role in flowering dynamics and be the target genes for yield breeding. Also, most of the genes responded to various abiotic stresses especially heat stresses, indicating that *GRFs* may serve in stress adaptation. 

### 2.7. Exploring the Expression Patterns of GRFs Target Genes under Diverse Conditions

To explore the expression patterns of the target genes of GRF family genes in Arabidopsis, we performed expression analysis of a selection of 109 common genes under diverse stresses and different developmental stages. The circular heatmap indicated that the expression levels in the shoot were much higher than that in the root (*p* = 4.41 × 10^−8^) under heat stress overall (Figure 8a; Appendix A). In addition, most of the genes in the shoot had higher expression values after 3 h of heat treatment, while the genes in the root were significantly induced after 12 h of heat treatment, indicating that different tissues responded to high temperature stress at different times, and leaves may sense the high temperature signal and then passed it on to the root. Among them, *ANAC029* (*AT1g9490*), *AP2/B3-like* (*AT4g33280*) and *REM19* (*AT3g06220*) were significantly induced under heat treatment after 1.0 h in the shoot, 3.0 h in the shoot and 3.0 h in the root, respectively (Figure 8a), indicating that these genes may act critical roles in response to heat stress regulated by GRFs. Previous papers reported that one of the AP2/B3-like gene members *REM16* acted as upstream regulators of suppressor of overexpression of constans (*SOC1*) and flowering locus T (*FT*) in flowering pathways [39]. Taken together, the 109 common genes of AtGRFs responded to high temperature to different degrees, suggesting that GRFs were involved in the heat stress in various pathways.

For further precise analysis, 25 out of 109 common genes encoding transcription factors belonging to different families in Arabidopsis with large expression differences were selected. All these genes could be divided into two groups (Figure 8b; Appendix A). Among them, *AT3g61890* (encoding a homeodomain leucine zipper class I protein, HB12), in cluster 1, had a higher expression level in flower stage 15 than other organs (Figure 8b). In cluster 2, four genes, *YABBY1* (*AT2g45190*), *AS1* (*AT2g37630*), *GRF1* (*AT2g22840*) and *DOF5.8* (*AT5g6940*) were highly expressed in the shoot apex, inflorescence, shoot apex & transition and shoot apex, vegetative, respectively (Figure 8b). All of these results suggested that GRFs involved in the control of cell differentiation and plant growth. In abiotic stresses, two groups could be separated based on their expression patterns (Figure 8c; Appendix A). Obviously, the gene expression levels in cluster 1 were significantly higher than that in cluster 2 (*p =* 8.28 × 10^−45^). Among them, *TGA1* (*AT5g65210*), *SVP* (*AT2g22540*), *ANAC083* (*AT5g13180*) and *WRKY70* (*AT3g56400*) had the top four expression levels in a variety of abiotic stresses. Interestingly, *WOX3* (*AT2g28610*) *and AP2/B3-like* (*AT4g33280*) in cluster 2, were significantly induced by cold stress after 12 h in root and by drought stress after 3 h in the shoot, respectively (Figure 8c). Taken together, GRFs were involved in a wide range of abiotic stresses, which provided the basis and guidance for studying the function of *GRFs* in different environments.

### 2.8. Phylogeny, Expression Level and Interactive Network Analysis of GRF and GIF Gene Family in Arabidopsis

As the interaction protein of GRF, GIF participates in plant growth and development and stress response cooperatively [16]. Three *GIF* gene members were identified in Arabidopsis (Appendix A). To further investigate the relationship between GRF and GIF, a polygenetic tree was constructed using the amino acid sequences of AtGRFs and AtGIFs. Expression patterns of *AtGRFs* and *AtGIFs* in various tissues were also detected. There was no significant difference between *AtGRFs* and *AtGIFs* in diverse tissues. Among the three *AtGIF* gene members, *AtGIF1* was highly expressed in flowers, while *AtGIF2* and *AtGIF3* had similar expression patterns in various tissues (Figure 9a; Appendix A). While most of the *AtGRF* genes were highly expressed in flowers, especially in *AtGRF5* and *AtGRF6.* (Figure 9a; Appendix A). The expression pattern of *AtGRF6* in the embryo and endosperm was second only to that in the flower, indicating GRF gene family may be involved in floral pathways. Furthermore, the expression differences of *AtGRFs* and *AtGIFs* under heat treatment were detected. *AtGRF6* had higher expression levels and reached the peak value at 12 h under heat treatment in the shoot, followed by *AtGRF8* at 1 h under heat treatment in root and *AtGIF2* at 3 h under heat treatment in the shoot (Appendix A). *AtGRF1, AtGRF2* and *AtGRF3* kept lower expression levels after heat treatment (Figure 9b; Appendix A). 

Based on expression analysis, the results indicated that most GIF family genes had a closer relationship with GRF family genes except *AtGIF3* (Figure 9c). Among them, *AtGIF1* had a strong correlation with *AtGRF5* and *AtGRF8*, and *AtGIF2* had a strong correlation with and *AtGRF1* and *AtGRF3,* indicating their tight interaction in playing roles (Figure 9c). Also, GRF gene family had a strong correlation with themselves, such as *AtGRF1* and *AtGRF2, AtGRF6* and *AtGRF7* and *AtGRF5* and *AtGRF8,* suggesting that these genes had evolved similar domains.

Further, we performed further interactive network analysis by calculating the Pearson correlation coefficients (PCCs) between these two gene families. There were 45 connections between any two genes in the network with PCC >0.6, and only 2 (4.44%) connections were identified with negative regulation (Figure 9d; Appendix A). Among them, *AtGRF1* and *AtGIF1* had the most positive connections with other genes, accounting for 21.4% and 3.57%, respectively. Taken together, the genes with more connections with other genes may play an important role in synergetic participation in Arabidopsis growth and development and stress response, such as heat stress.

## 3. Discussion

GRF family transcription factors are involved in regulating various life processes during plant growth and development by controlling cell division [12,23]. In order to study the evolutionary developmental relationships of the GRF gene family in seven species, we performed a comprehensive analysis of GRF gene family with 70 sequences in seven species in this study. Gene structure, classification, evolutionary, duplication, functional analysis, interaction network and expression patterns of this gene family were analyzed. Our results will pave the way for studies of the functions of GRF gene family in terrestrial plants and will further our understanding of this gene family in other plants, especially the bioinformatic information and functional reference will be useful for future comparative and functional genomic studies of the GRF gene family.

Previous studies reported that GRF proteins were characterized by two conserved motifs, QLQ and WRC [10,12]. In this study, QLQ domians, WRC domains, Rmic domains were also detected in several GRF sequences of representative plants. Previous reports indicated that all 19 *ZmJMJ* genes were responsive to heat stress treatment [25], *AtJMJ30/JMJ32*-mediated histone demethylation regulated flowering control at warm temperature [40], and *AtJMJ13* acted as a temperature- and photoperiod-dependent flowering repressor [41], suggesting their potential roles in heat stress response [25,27]. Interestingly, the domain ATT-l was detected in *Maploy0090s0035.1* in *M. polymorpha* (Figure 1a⑧). ATT-1 promoter was reported to have great activity in the inner integument [42], indicating it was associated with the inner seed coat development. Therefore, *GRFs* researchers may consider carrying out experiments under abiotic stress. 

Previous studies have indicated that the GRF gene family is involved in various hormone signaling transduction pathways. GRFs are involved in the brassinosteroid (BR) pathway in *A. thaliana* and gibberellin (GA) biosynthesis in *Nicotiana tabacum* [43]. In this study, many hormone-related cis-elements in promoters such as gibberellin-responsive element, MeJA-responsiveness element, and salicylic acid responsiveness element were detected (Figure 2), which suggested GRFs were regulated by various hormones. Many stress response elements in the cis-acting elements of seven representative plants such as drought-inducibility element and low-temperature responsiveness element were also detected (Figure 2), which was consistent with the fact that the GRF gene family responds to stress responses [23]. Considering the diversity of function and distribution of cis-elements in the promoter regions, we speculate that GRF gene family may differentially regulate the expression of genes that are involved in plant development under abiotic stresses. Therefore, it is highly important to further study the functional characterization of GRF gene family under stresses and attain new insights into the molecular mechanism.

Gene duplication plays a pivotal role in gene family evolution, speciation and diversification of plants, which is the primary mechanism for the generation of novel evolutionary innovation to improve the adaptability in unfavorable environments [44]. In this study, the number of gene loss was more than gene duplication in seven representative plants, and the dispersed and WGD/segmental were the main factors in GRF gene family expansion (Figure 3). This suggested that WGD/segmental played an important role in the GRF gene duplication. Dispersed duplication and WGD/segmental duplication effectively promoted the expansion of the GRF gene family, but tandem duplication events were not detected (Figure 3), which has been reported previously [45]. Furthermore, the higher plants had more orthologous gene pairs than the lower plants, even some lower plants did not have orthologous genetic pairs, such as *O. lucimarinus* and *M. polymorpha* (Appendix A). It indicated that the higher plants existed the genome-wide polyploidy events. In most species, the Ka/Ks ratio < 1, indicating that purifying selection accounted for a large share. 

GRF genes were highly expressed in growing and developing tissues, and weakly expressed in mature stem and leaf tissues [12]. Consistent with previous research results, GRFs were mainly expressed in the shoot apex and flowers compared with other organs in Arabidopsis and rice (Figure 7). Floral bud development in the flowering stage was greatly associated with fruit yield, quality and cultivation. The induction and development of flower buds could be regulated by several floral genes and environmental factors. The high expression patterns of most GRFs in the flower stage indicated that they may have functions in flower bud development and fruit production improving through responding to environmental factors and associating with other floral genes. The role of GRF gene family regulating flowering initiation and development has been reported [7]. *OsGRF1* controls flowering time and *OsGRF6* is involved in floral organ development in rice [46]. In addition to high expression levels in flower stages, comparatively higher expression patterns were found in the meristem, indicating their potential functions in organ formation and cell division. Studies claimed GRFs participating in abiotic stress were accumulated recently. Overexpression of *ZmGRF10* in maize led to a reduction in leaf size and plant height [25], GRF gene family in tomato responded to abiotic (NaCl, heat, cold, drought) stress, which could be favorable evidence to our results. The activation of stress-responsive genes increases plant tolerance to overcome unfavorable environments. Therefore, we also studied the expression patterns of GRF family genes under various abiotic stresses and found that the expression level of most GRF genes was up-regulated under stress treatment (cold and heat), which suggested that GRF gene family have wide responses to abiotic stress. Our expression analysis may provide a solid foundation for future studies of GRF gene family regulatory functions during flowering development and under abiotic stresses.

Protein interaction networks composed of proteins that interact with one another help to find candidate genes that participate in multiple aspects of life processes such as biological signaling, gene expression regulation, substance metabolism and cell cycle regulation. To further explore the potential functions of GRFs, we constructed an interaction network in Arabidopsis with 6689 gene pairs and performed functional enrichment analysis (Figure 5a). We found that 120 common genes encoded transcription factors belonging to 25 gene families and the target genes were identified to interact with other transcription factors, such as MYB, HB, WRKY and bHLH. The results of this analysis further corroborate a previous study that noted that TaGRF6-A positively regulated salt stress tolerance by interacting with TaMYB64 [47]. A further example noted here also showed that OsGRF14e interacts with OsCPK21 to promote the salt stress response in rice [48].

Function analysis for GRF target genes showed GRF gene family was involved in multiple pathways and mainly focused on metabolism and abiotic stresses, which is consistent with our previous speculation that GRF gene family coordinates plant growth with stress responses [12,23] and serves as a useful resource for future study on the GRFs molecular functions. GRFs functions were reflected by regulating the expression of target genes. The expression patterns of common genes in different tissues and response to different stresses are also worth studying. We performed expression analysis of selected 109 common genes under diverse stresses and developmental stages (Figure 8). Among them, the expression levels of *ANAC029*, *AP2/B3-like* and *REM19* significantly increased after heat treatment in different tissues, while most of the genes were highly expressed in the shoot apex and flower stages. It is reported that *BnaYAB2*, *BnaYAB3*, and *BnaYAB5* in *B. napus* were expressed in various tissues at the developmental and flowering stage [49]. High levels of *AtAS1* expression were found in tissues with highly proliferative cells [50], suggesting that the GRF gene family may act critical roles in response to heat stress and floral bud development. Besides, the four genes *TGA1*, *SVP*, *ANAC083* and *WRKY70* were most responsive to abiotic stresses. Previous reports also indicated that interactions between *SVP* and *FLM* isoforms modulated the temperature-responsive induction of flowering in Arabidopsis [51]. *StTGAs* were generally expressed in different tissues at different stages of development [52]. At *WRKY70* was involved in drought response [53], and *IbNAC3* modulates combined salt and drought stresses [54], while the expression level of *TaWRKY70* was increased significantly when exposed to high temperature. 

At present, evidence is accumulated that GRFs integrate environmental stress signals with growth programs to help balance stress responses during growth stages [12,23]. Expression analysis of *AtGRFs* and *AtGIFs* under various abiotic stresses and in different tissues also confirmed the above conclusion (Figure 9). Furthermore, we constructed a network between GRFs and GIFs using expression data of various tissues in Arabidopsis. The results indicated that 12 out of 45 connections between any two genes in the network were detected with PCC > 0.8 and well suggested the co-regulatory relationship between GRF and GIF, which lays a foundation for further experimental researches on the interaction. Previous reports indicated that GRF4 and its cofactor GIF1 increased the efficiency and speed of regeneration in wheat, triticale and rice [55]. The simultaneous increases of *AtGRF3* and *AtGIF1* promoted the development of larger leaf sizes than increased individually [20].

## 4. Materials and Methods

### 4.1. GRF Gene Family Identification and Physicochemical Properties

Feature domains QLQ (PF08880) and WRC (PF08879) of GRF proteins were used to identify GRF candidates on the Pfam database with an e-value < 1 × 10^−4^. To ensure the members’ accuracy, SMART and CDD databases were used to detect domains [56]. The sequences of these GRF of examined plants in Fasta format were downloaded from the phytozome database (https://phytozome-next.jgj.doe.gov/blast-search, accessed on 19 June 2022). The protein physicochemical properties (amino acid number, molecular weight and isoelectric point) of these GRF proteins were subsequently obtained using ExPASy website (http://web.expasy.org/, accessed on 22 July 2022). Gene structures of GRFs were displayed using the online website GSDS (http://gsds.cbi.pku.edu.cn/, accessed on 30 July 2022). Distribution of GRF gene members in different species were mapped using software Tbtools. The GRF gene number, average length, WRC and QLQ domain numbers were calculated using Origin 2022 software. 

### 4.2. Phylogenetic Tree Construction and Conserved Motif Comparison

The multiple GRF protein sequences were combined into one in single species by using DNAMAN software. Then, the protein sequences were aligned by Mafft v7.471 software with maxiterate at 1000 [57]. The maximum likelihood tree was constructed using MEGA7 software. The phylogenetic trees of representative plants were made using the iTOL website (https://itol.embl.de/, accessed on 15 August 2022 [58]. 

The amino acid sequences of GRF genes in seven representative plants were submitted to MEME (https://meme-suite.org/meme/tools/meme, accessed on 18 August 2022) with default parameters except for the maximum number of motifs was set 10 [59]. The 10 motif sequences were used as queries to identify the loss and retention.

### 4.3. Identification and Visualization of Cis-Acting Elements in the Promoters of GRF Genes

Tbtools was used to extract the 2 kb upstream of the translation initiation site of each GRF family gene in seven representative plants. Then, the sequences were submitted to the online website PlantCARE (http://bioinformatics.psb.urgent.be/webtools/plantcare/html, accessed on 20 August 2022) for cis-acting elements prediction [60]. Besides, the number and distribution of elements in seven representative plants were mapped using the inner script with R software. The 2 kb upstream sequences of seven representative plants were aligned by MEGA7 software with neighbor-joining method to build phylogenetic tree using the Poisson model with 1000 bootstrap replications.

### 4.4. Duplication and Loss Detection for GRF Genes

The duplication and loss of GRF family genes were detected by Notung2.9 software [61]. The evolutionary timescale of life in seven representative plants were calculated by the online website TIMETREE5 (http://www.timetree.org/, accessed on 25 August 2022). All the information on the trees was illustrated by Ai software.

### 4.5. Syntenic GRF Genes Recognition

McscanX was used for analyzing the syntenic GRF genes among the seven representative plants and among main inter-species classified by genus [62]. Ka/Ks between the main representative plants were calculated by syntenic GRF genes with Tbtool sotware and JCVI v0.9.14 (https://pypi.org/project/jcvi, accessed on 29 August 2022) set default parameters.

### 4.6. Interaction Network Construction of GRF Gene Family

The interactive genes of GRF gene family in Arabidopsis were detected by the integrated gene regulatory network (iGRN) database (http://bioinformatics.psb.ugent.be/webtools/iGRN/, accessed on 5 September 2022) with score ≥ 0.60 [63]. The interactive genes could be divided into upstream genes and downstream genes. Then, the interaction network between GRF family genes and target genes was constructed using Gephi software (v0.92) with default parameters [64].

### 4.7. Functional Enrichment Analysis of Main Target Genes

To predict the function of target genes, several online websites SwissProt (http://www.uniprot.org, accessed on 8 September 2022), GO, KEGG (http://www.genome.jp/kegg/, accessed on 12 September 2022), Unirpot (https://www.uniprot.org/, accessed on 12 September 2022) [65] and InterPro (https://www.ebi.ac.uk/interpro/, accessed on 15 September 2022 [66] were used to annotated GRF target genes. Then, the useful functional enrichment with *q*-value < 0.05 were submitted into the online website Omicshare (https://www.omicshare.com/tools/Home/Soft/getsoft, accessed on 17 September 2022) to illustrate the function and relative gene numbers.

### 4.8. Expression Patterns of GRF and GRF-Regulated Genes in Different Tissues and under Diverse Conditions

The expression data of 25 typical transcription factors of the common downstream- and upstream- genes were downloaded from the website of the Arabidopsis eFP browser (http://www.bar.utoronto.ca, accessed on 18 September 2022). The expression data of GRF genes in maize, soybean and rice were extracted on the website (http://ipf.sustech.edu.cn/pub/athrna/, accessed on 20th September 2022). The heatmap was demonstrated by Tbtool software and the online Bioinformatics website (http://www.bioinformatics.com.cn/, accessed on 25 September 2022) [67].

To further explore the potential functions of GRF genes in response to heat stress and interaction with GIF genes in Arabidopsis, transcriptome data from tair website (https://www.arabidopsis.org/, accessed on 27 September 2022)after high temperature treatment (0 h, 0.5 h, 1 h, 3 h, 6 h, 12 h and 24 h) was used to detect the expression levels of GRF genes in Arabidopsis. In addition, transcriptome data including root, stem, leaf, flower, seed, endosperm and embryo were used to analyze their roles of GRF genes in different tissues.

### 4.9. GRF-GIF Interaction Network Construction

The protein sequence of the GRF genes and GIF genes of Arabidopsis downloaded in tair database database (https://www.arabidopsis.org/, accessed on 8 October 2022) [68]. PCCs between GRF and GIF were calculated by SPSS software according to the GRF and GIF expression value under heat stress. PCC > 0.6 or PCC < −0.6 were respectively represents the positive and negative regulation. The interactive relationship were illustrated using Gephi software.

## Figures and Tables

**Figure 1 plants-12-02790-f001:**
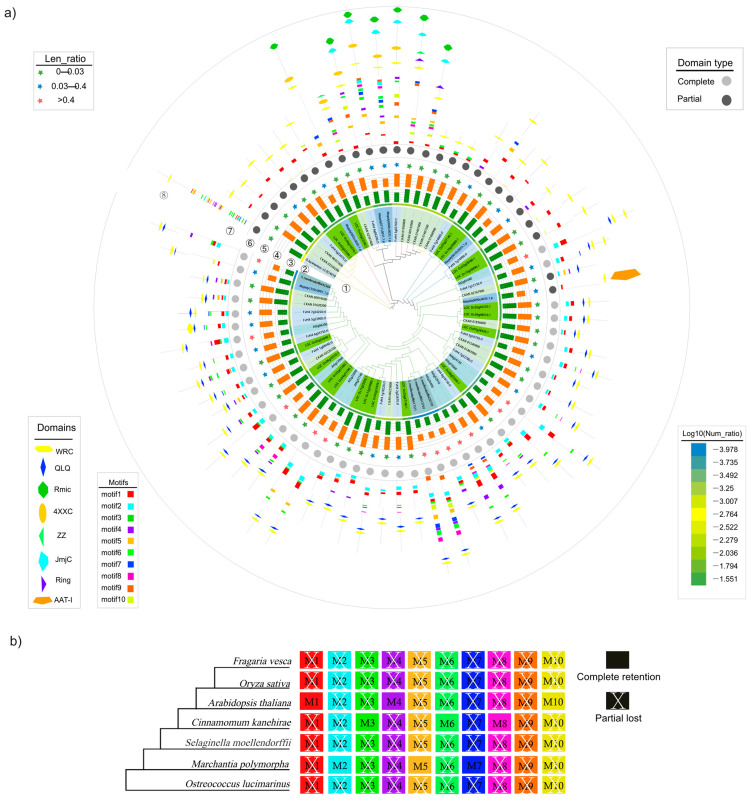
Phylogenetic, conserved motif, duplication type, and evolutionary trajectory analyses of GRF family genes from seven representative plants. (**a**) Phylogenetic tree of seven representative plants (① A Phylogenetic Maximum-likelihood tree were generated based on with amino acid sequences of GRF gene family. ② Log10 number ratio of GRF family genes compared with whole-genome genes in each species. ③ Log10 number of whole-genome genes in each species. ④ Log2 number of GRF family genes in each species. ⑤ Length ratio of GRF family genes compared with whole-genome genes in each species. ⑥ The Completeness of the domains in representative plants, solid means WRCs and QLQs, hollow means only one of them. ⑦ The motifs of representative plants. ⑧ The symbol domains of representative plants). (**b**) Motif loss and retention in seven representative plants. The white X indicates that the motif was lost in some genes.

**Figure 2 plants-12-02790-f002:**
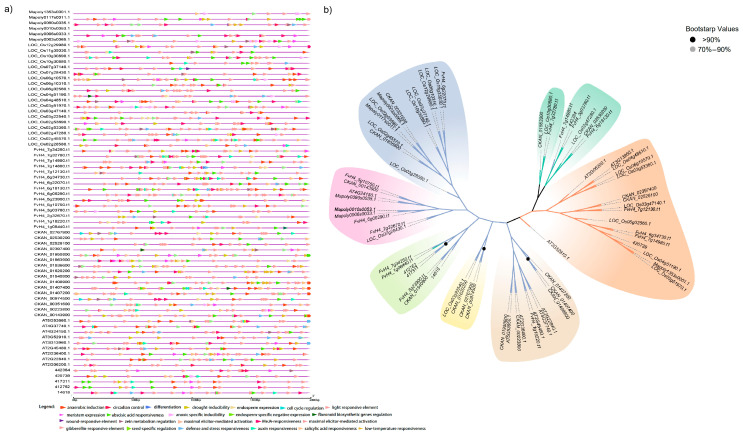
**The cis-acting elements in the promoter sequences of the GRF gene family.** (**a**) The cis-acting elements in the promoter of each GRF family gene. (**b**) A phylogenetic tree constructed using 2 kb upstream sequence of promoters in seven representative plants.

**Figure 3 plants-12-02790-f003:**
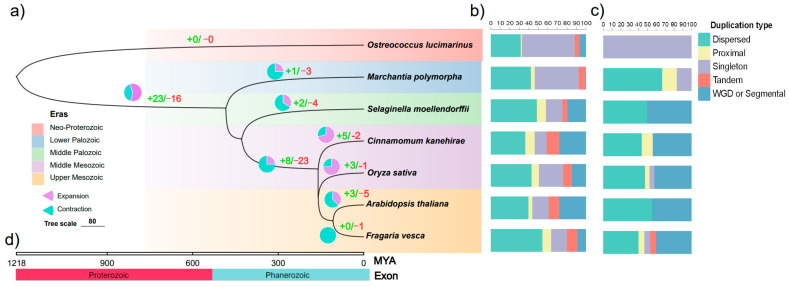
Phylogenetic tree of seven representative plants based on evolutionary time. (**a**) The pie graph represents the proportion of gene families that underwent duplication (green) or loss (red) when comparing with their most recent common ancestor. (**b**) The proportion of each duplication type for all genes. (**c**) The proportion of each duplication type for GRF family genes. (**d**) Estimated divergence times and the time scale are shown at the bottom.

**Figure 4 plants-12-02790-f004:**
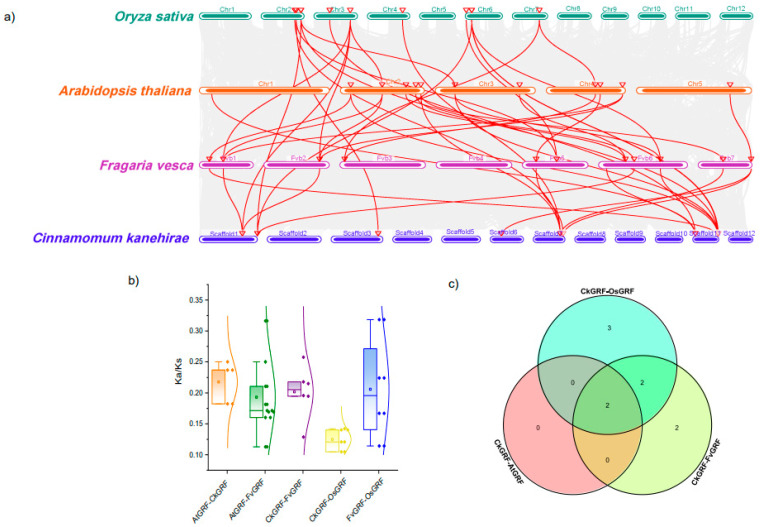
Syntenic GRF genes among inter- and intra- representative plants. (**a**) Syntenic GRF genes distribution in representative plants. (**b**) Ka/Ks value of GRF genes box diagram in representative plants. (**c**) Venn diagram analysis of syntenic GRF genes between representative plants. different colors represent different species.

**Figure 5 plants-12-02790-f005:**
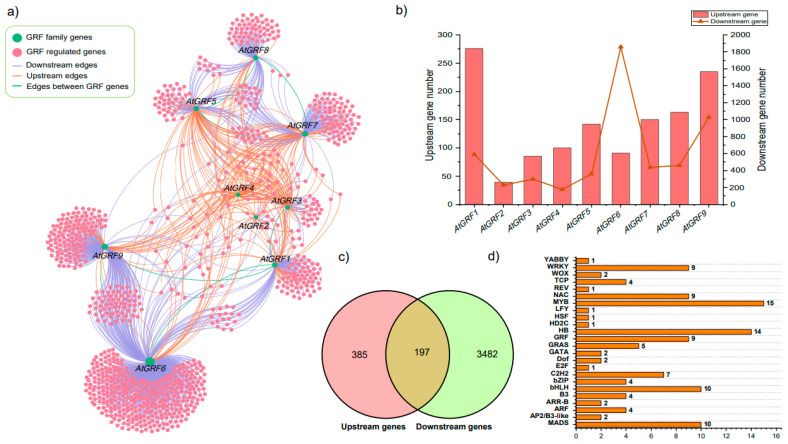
The interaction network among GRF family genes, and their upstream and downstream-regulated genes of GRF family genes in Arabidopsis. (**a**) GRF genes network with regulated genes. (**b**) Number of upstream and downstream genes for each GRF family gene in Arabidopsis. (**c**) Specific and shared genes between downstream and upstream genes. (**d**) The type and number of transcription factors in upstream- and downstream-regulated genes.

**Figure 6 plants-12-02790-f006:**
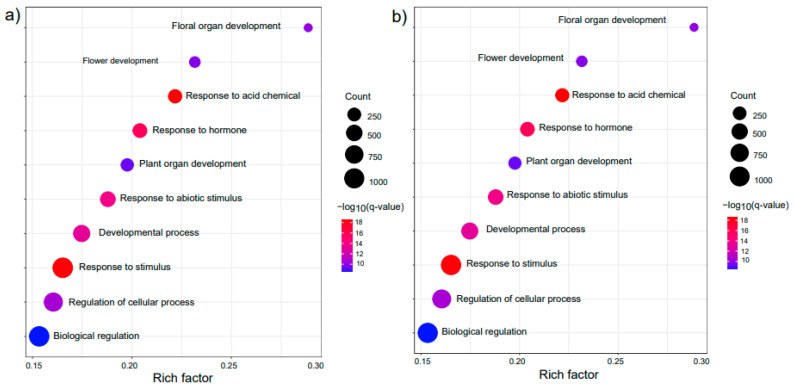
Expression of *GRFs* in soybean and maize under different stresses and development stages. (**a**) Expression heatmap of *GRFs* in soybean under various abiotic stresses and different tissues. (**b**) Expression heatmap of *GRFs* in maize under various abiotic stresses and different tissues.

**Figure 7 plants-12-02790-f007:**
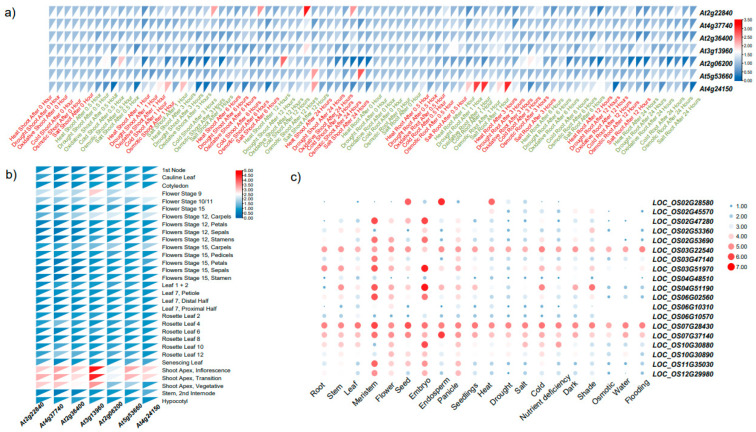
Expression of *GRFs* in Arabidopsis under different stresses and development stages. (**a**) Expression heatmap of *GRFs* in Arabidopsis under different treatments. (**b**) Expression heatmap of *GRFs* in Arabidopsis at various developmental stages in different tissues. (**c**) Expression heatmap of *GRFs* in rice in different tissues and under different treatments.

**Figure 8 plants-12-02790-f008:**
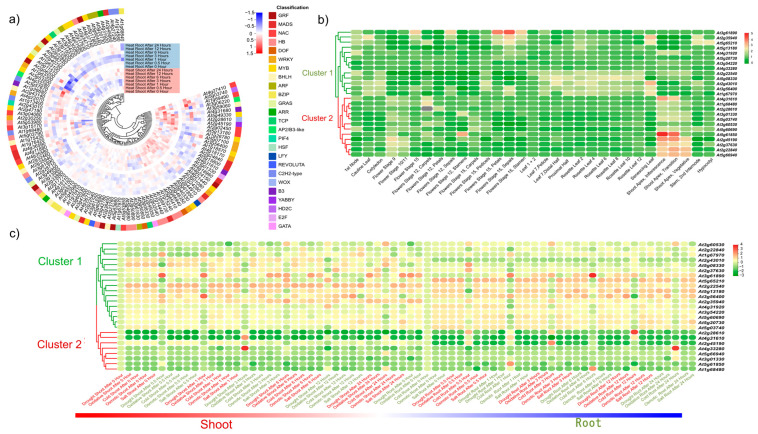
**Expression of selected *GRFs* under different stresses and development stages.** (**a**) Expression heatmap of selected transcription factors under heat treatment for different lengths of time. (**b**) Expression heatmap of selected genes at various developmental stages in different tissues. (**c**) Expression heatmap of selected genes under various abiotic stresses.

**Figure 9 plants-12-02790-f009:**
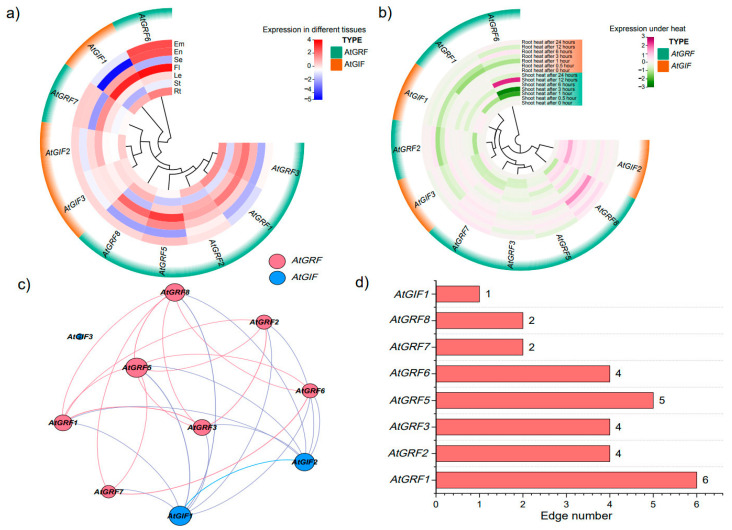
Construction of *GRFs* and GIFs interactive network. (**a**) Expression heatmap of *AtGRFs* and *AtGIFs* under heat treatment in shoot and root. (**b**) Expression heatmap of *AtGRFs* and *AtGIFs* in different tissues. (**c**) Interaction network of GRF and GIF gene families with the PCC > 0.6. (**d**) Number of edges made by *GRFs* and *GIFs*.

## Data Availability

Date is contained within the article and Appendix A.

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
