# Peer review of "Comparatively Evolution and Expression Analysis of GRF Transcription Factor Genes in Seven Plant Species"

_plants, 2023, doi:10.3390/plants12152790_

Round 1
Reviewer 1 Report (Previous Reviewer 2)
I think the manuscript should be improved however for the publication.
1. “gene” g and“seven”s should be capitalized in the title.
2. line 5, Is that an extra "and" at the end?
3. line 52 and 69, GIF and CKX, Full name at first use. Check manuscript and correct all. Such as DREB2A, PI and so on. I am not going through your whole brief to correct this lapse. it is on you to attend to this throughout your manuscript.
4. Line 85, I can't find any reason in manuscript to explain why chose these 7 species. you should explain in a few words.
5. line 133, Figure 1 a little blurry and heading should be bold.
6. line 189, should be in a consistent format as above. Do need a space after the (b)? Check manuscript and correct all. Such as line 139.
7. Supplementary Materials section should be added. There are also some typographical errors You have to revise.
Author Response
Dear reviewer:
Thank you very much for your comments and professional advice. These comments are all valuable and very helpful for revising and improving our paper. Based on your suggestion and request, we have revised the details. Meanwhile, the manuscript had be reviewed and corrected the text format. We hope that our work can be improved again. Here are some details:
Response to Reviewer 1 Comments
Point 1:“gene” g and“seven”s should be capitalized in the title.
Response 1: Thanks for your careful advice, we have corrected the capital letters.
Point 2: line 5, Is that an extra "and" at the end?
Response 2: We feel sorry for our carelessness, and we have deleted the extra “and”.
Point 3: line 52 and 69, GIF and CKX, Full name at first use. Check manuscript and correct all. Such as DREB2A, PI and so on. I am not going through your whole brief to correct this lapse. it is on you to attend to this throughout your manuscript.
Response 3: We thank the reviewer for pointing this out, and we have already added full name at first use. Line 52 GRF-Interacting Factor (GIFs), line 53 Switch/Sucrose Non-fermenting gene (SWI2/SNF2), line 71cytokinin oxidase/dehydrogenase genes (CKX), line 76 DEHYDRATION RESPONSIVE ELEMENT-BINDING PROTEIN2A (DREB2A), line 341 suppressor of overexpression of constans (SOC1) and flowering locus T (FT).
Point 4: Line 85, I can't find any reason in manuscript to explain why chose these 7 species. you should explain in a few words.
Response 4: Thank you for your valuable advice. In order to reveal the evolutionary relationship and expression pattern of GRFs, we selected these seven representative species according to the evolutionary order Thallophyte, Bryophyta, Pteridophyta, Monocots, Lauraceae, Rosales and Cruciferae, and we have added few words on line 100.
Point 5: line 133, Figure 1 a little blurry and heading should be bold.
Response 4: Thank you for your careful check. We have replaced the figure 1 with a clearer one, added the vector image, and bolded the heading.
Point 6: line 189, should be in a consistent format as above. Do need a space after the (b)? Check manuscript and correct all. Such as line 139.
Response 6: Thank you very much for your reminding. We have checked again and unified the format, and we understand your comments as follows: in the body, there is no space after (Figure 1b) with punctuation, while a space is added after (b) in the note.
Point 7: Supplementary Materials section should be added. There are also some typographical errors You have to revise.
Response 7: Thanks for your careful advice, we have added supplementary tables and revised the typographical errors.
Reviewer 2 Report (Previous Reviewer 3)
I suggest accepted the manuscript.
Author Response
Thanks for your reply.
This manuscript is a resubmission of an earlier submission. The following is a list of the peer review reports and author responses from that submission.
Round 1
Reviewer 1 Report
The Authors provided an extensive bioinformatical analysis of GRF family in different plant species. The results obtained could be useful in further studies. However, I have some questions concerning the study design and its realization.
The Authors have divided 69 plant species into seven group, one of them is “Others” (31 species). Why did you combine all these species in one group? According to the list, the group contains at least one monocot plant (Ananas comosus) and one gymnospermous plant (Thuja plicata).
In the title and article’s abstract analysis of 69 plant species was declared. In fact, the authors did not analyze them all. For example, 69 species was ranked and only 10 top and 10 bottom species were analyzed or for further analysis one specie from each category was randomly selected (only 7 species). Are you sure that data obtained can be extended to all species under investigation?
The Figures are redundant and its quite difficult to understand it by human eyes. Probably you should reduce them and show only the key information and results.
English Language may be improved, but its not necessary
Author Response
Dear Reviewer,
Thank you very much for your comments and professional advice. These comments are all valuable and very helpful for revising and improving our paper, as well as the important guiding significance to our researches. Based on your suggestion and request, we have made corrected modifications on the revised manuscript. The responses in detail are as follows:
Point 1: The Authors have divided 69 plant species into seven group, one of them is “Others” (31 species). Why did you combine all these species in one group? According to the list, the group contains at least one monocot plant (Ananas comosus) and one gymnospermous plant (Thuja plicata).
Response 1: Thank you very much for pointing out this important issue. Different taxonomies have different species groups. In this study, we classified species according to family genus, and aimed to explore the expression patterns under diverse conditions. This classification did not affect the representative species we selected, and we mainly focused on the selected species. In subsequent studies, we will refine the species types and make the classification more reasonable.
Point 2: In the title and article’s abstract analysis of 69 plant species was declared. In fact, the authors did not analyze them all. For example, 69 species was ranked and only 10 top and 10 bottom species were analyzed or for further analysis one specie from each category was randomly selected (only 7 species). Are you sure that data obtained can be extended to all species under investigation?
Response 2: Indeed, the analysis of several species do not represent all the plants. With the large amount of expression and sequence data of a large number of species, we tried to find convenient algorithms to process and reveal the evolution laws of lower and higher plants, but we failed due to inappropriate algorithms or abundant sequences. Fortunately, an evolutionary article published in Horticulture research gave us the inspiration (https://doi.org/10.1093/hr/uhac035). In this study, authors used the top 10 and bottom 10 species, as well as representative species, to reveal evolutionary laws, which indicating that this is a convenient and convincing way.
Point 3: The Figures are redundant and its quite difficult to understand it by human eyes. Probably you should reduce them and show only the key information and results.
Response 3: Thanks for your suggestion, we have moved Figures 4 and 8 into supplementary materials.
Reviewer 2 Report
The manuscript reported “Comparatively Evolution and Expression Analysis of GRF 2 Transcription Factor genes in 69 Plants Species”. It gave us interesting ideas, clear results, logical discussion, and adequate conclusion. Before publication, many details should be revised. The comments are appended below.
1. You should check the italic names of genes and plant Latin names, such as line 315 to line 319.
2. GIF and CKX, and so on, should give full name at first use, such as line 56 and 73. You should check manuscript and correct all.
3. You should capitalize Gene in title, delete and in line 5, delete underline in line 136, and so on.
4. 2.9 The title is …Under Diverse Conditions in line 425, but you only introduced the heat treatment. The title and the content are mismatching. You can change the title “under heat donditions”.
5. The results of 2.8, 2.9 and 2.10 all from transcriptome or website prediction. You have no experimental data regardless of expression or interaction. You are suggested to do Quantitative analysis or Interaction verification of several major genes.
In short, you should check the typographical errors, English correctness, References carefully.
Regardless, I really enjoyed your discovery and take my hat off to your labs and this huge work! Congratulations!
Author Response
Dear reviewer:
Thank you very much for your comments and professional advice. These comments are all valuable and very helpful for revising and improving our paper, as well as the important guiding significance to our researches. The responses in details are as followed:
Point 1: You should check the italic names of genes and plant Latin names, such as line 315 to line 319.
Response 1: Thanks for your careful advice, we have checked and corrected again for the format of genes and Latin names in our new MS.
Point 2: GIF and CKX, and so on, should give full name at first use, such as line 56 and 73. You should check manuscript and correct all.
Response 2: Thank you for your careful check, we have already added full name at first use.
Point 3: You should capitalize Gene in title, delete and in line 5, delete underline in line 136, and so on.
Response 3: We feel sorry for our carelessness. These mistakes have been revised in our new MS.
Point 4: 2.9 The title is …Under Diverse Conditions in line 425, but you only introduced the heat treatment. The title and the content are mismatching. You can change the title “under heat conditions”.
Response 4: We sincerely thanks for your careful remainder. As suggested by the reviewer, we have corrected the “diverse conditions” into “heat conditions” in our new MS.
Point 5: The results of 2.8, 2.9 and 2.10 all from transcriptome or website prediction. You have no experimental data regardless of expression or interaction. You are suggested to do Quantitative analysis or Interaction verification of several major genes.
Response 5: Indeed, it will be more convincing if we do quantitative analysis or interaction verification. However, this manuscript is a preliminary study focusing on the bioinformatics level and that will be further developed in subsequent studies. In addition, it might cause unreliable results in the screening of key genes from thousands of genes in diverse species. Together with the difficult sample collection and processing conditions for certain lower plants such as flagellates, it might be difficult to carry out.
Reviewer 3 Report
Growth regulatory factors (GRF) are plant-specific transcription factors that play pivotal 21 roles in growth and various abiotic stresses regulation. The manuscript investigated the adaptive evolution of GRF gene 22 family in land plants. The manuscript is generally well-written and the investigation is scientifically sound. I suggest accepting the manuscript with minor revision.
1. Line 5 delete and.
2.Figures 1, 3, 4, 10, 11 are not clear, could you change them to vectorgraph.
3. The text is well-written, there are numerous errors in the references.
4. A large number of species names in the main text are not italicize.
Author Response
Dear reviewer:
Thank you very much for your comments and professional advice. These comments are all valuable and very helpful for revising and improving our paper, as well as the important guiding significance to our researches. The response in detail are as folllowed:
Point 1: Line 5 delete and.
Response 1: We feel so sorry for our mistake, and we have deleted “and” in line 5 in our new MS.
Point 2: Figures 1, 3, 4, 10, 11 are not clear, could you change them to vectorgraph.
Response 2: Thank you for your helpful suggestions, we have added and uploaded all the figures in pdf format for clear reading.
Point 3: The text is well-written, there are numerous errors in the references.
Response 3: So sorry for our carelessness, we have rechecked the references and made changes in our new MS.
Point 4: A large number of species names in the main text are not italicize.
Response 4: Thank you for your kind suggestions, we have rechecked the species name and italicized the text.
Round 2
Reviewer 1 Report
“In this study, we classified species according to family genus, and aimed to explore the expression patterns under diverse conditions. This classification did not affect the representative species we selected, and we mainly focused on the selected species.”
In the article the evolutionary and expression analysis of GRF gene family from 69 representative species was declared. If the main aim of the study was to explore expression patterns of some concrete species, you should write an article about it. It will be an article containing less data, but more accurate. Classification of species where almost half were included in “Others” is extremely confusing. You should revise the experiment design and probably divide this manuscript into two articles. But now, unfortunately, I have to reject the article.